# Varidnaviruses in the Human Gut: A Major Expansion of the Order *Vinavirales*

**DOI:** 10.3390/v14091842

**Published:** 2022-08-23

**Authors:** Natalya Yutin, Mike Rayko, Dmitry Antipov, Pascal Mutz, Yuri I. Wolf, Mart Krupovic, Eugene V. Koonin

**Affiliations:** 1National Center for Biotechnology Information, National Library of Medicine, Bethesda, MD 20894, USA; 2Center for Algorithmic Biotechnology, Institute for Translational Biomedicine, St. Petersburg State University, 199004 St. Petersburg, Russia; 3Archaeal Virology Unit, Institut Pasteur, Université Paris Cité, CNRS UMR6047, F-75015 Paris, France

**Keywords:** human gut metagenome, double jelly roll capsid, *Varidnaviria*, *Vinavirales*, *Corticoviridae*, *Autolykiviridae*

## Abstract

Bacteriophages play key roles in the dynamics of the human microbiome. By far the most abundant components of the human gut virome are tailed bacteriophages of the realm *Duplodnaviria*, in particular, crAss-like phages. However, apart from duplodnaviruses, the gut virome has not been dissected in detail. Here we report a comprehensive census of a minor component of the gut virome, the tailless bacteriophages of the realm *Varidnaviria*. Tailless phages are primarily represented in the gut by prophages, that are mostly integrated in genomes of *Alphaproteobacteria* and *Verrucomicrobia* and belong to the order *Vinavirales,* which currently consists of the families *Corticoviridae* and *Autolykiviridae.* Phylogenetic analysis of the major capsid proteins (MCP) suggests that at least three new families should be established within *Vinavirales* to accommodate the diversity of prophages from the human gut virome. Previously, only the MCP and packaging ATPase genes were reported as conserved core genes of *Vinavirales*. Here we report an extended core set of 12 proteins, including MCP, packaging ATPase, and previously undetected lysis enzymes, that are shared by most of these viruses. We further demonstrate that replication system components are frequently replaced in the genomes of *Vinavirales*, suggestive of selective pressure for escape from yet unknown host defenses or avoidance of incompatibility with coinfecting related viruses. The results of this analysis show that, in a sharp contrast to marine viromes, varidnaviruses are a minor component of the human gut virome. Moreover, they are primarily represented by prophages, as indicated by the analysis of the flanking genes, suggesting that there are few, if any, lytic varidnavirus infections in the gut at any given time. These findings complement the existing knowledge of the human gut virome by exploring a group of viruses that has been virtually overlooked in previous work.

## 1. Introduction

In the recently adopted megataxonomy of viruses, bacteriophages with double-stranded (ds) DNA genomes belong to two vast realms, *Duplodnaviria* and *Varidnaviria* [1]. The viruses in these realms possess unrelated structural modules, including the distinct major capsid proteins (MCP) and packaging ATPases. The realm *Duplodnaviria* includes tailed bacteriophages and archaeal viruses, along with the related herpesviruses. The realm *Varidnaviria* consists of tailless bacteriophages, related archaeal viruses and numerous viruses of eukaryotes, in particular, those of the phylum *Nucleocytoviricota*, which includes giant viruses such as mimiviruses.

The great majority of cultivated bacteriophages are tailed and belong to the realm *Duplodnaviria*. Surprisingly, however, metagenomic analyses present a different picture, indicating that non-tailed varidnaviruses numerically dominate the upper oceans, comprising 50% to 90% of the detected virus particles [2,3]. The hallmark of the realm *Varidnaviria* is the double jelly-roll (DJR) MCP that forms characteristic icosahedral capsids [4,5]. The sequences of the DJR MCPs are highly divergent, and in some viruses beyond easy recognition, but the core structure is conserved across the entire diversity of the members of *Varidnaviria* infecting hosts from all three domains of life [6,7]. A search of metagenomic sequence data for contigs encoding DJR MCP using sensitive profile-based methods resulted in the discovery of an unexpected, broad variety of varidnaviruses, most likely infecting bacterial and archaeal hosts, that by far exceeded the known diversity of viruses of prokaryotes in the families *Turriviridae*, *Tectiviridae*, *Corticoviridae* and *Autolykiviridae* in the class *Tectiliviricetes* [8]. The most abundant group of these predicted tailless phages was ‘PM2-like’, which included numerous prophages. The isolated and sequenced members of this group are two phages infecting *Pseudoalteromonas* species, the eponymous PM2 [9] and Cr39582 [10] that belong to the family *Corticoviridae* [11], 10 Vibrio phages in the family *Autolykiviridae* [3], and the unassigned fNo16 phage [12].

The families *Corticoviridae* and *Autolykiviridae* jointly comprise the order *Vinavirales* [1]. A typical phage of this order has a 9-12 kb-long genome, which is either circular (*Corticoviridae*) or contains inverted terminal repeats (*Autolykiviridae*). The genomes of viruses in the order *Vinavirales* encode structural components of the virion as well as genes involved in genome replication, transcription regulation, and cell lysis [8,13]. It has been shown that proviruses related to members of *Vinavirales* are abundantly integrated in the genomes of aquatic bacteria [13], but their presence in other ecosystems has not been explored. Apart from DJR MCP and packaging ATPase, no proteins shared by all or most members of *Vinavirales* have been identified [13].

The human gut virome that plays a major role in microbiome dynamics has been extensively studied by metagenomics approaches. The overwhelming majority of the detected viruses are tailed bacteriophages of the realm *Duplodnaviria* [14,15,16,17]. In particular, the most abundant human-associated viruses belong to the group of crAss-like phages, recently classified by the ICTV into the order *Crassvirales*, that were originally discovered by metagenomics analysis and subsequently shown to include numerous, highly diverse members [18,19,20,21,22]. Several additional, new families of abundant tailed bacteriophages have been discovered by gut metagenomics as well [23,24].

In contrast, members of *Varidnaviria* apparently constitute a minor fraction of the human gut virome and remain poorly characterized. In this work, we screened 23,119 gut metagenomes (retrieved on March 2021) for DJR MCP and found that a substantial majority of the sequences encoding this hallmark protein of *Varidnaviria* belong to prophages that are distantly related to the families *Corticoviridae* and *Autolykiviridae* and appear to represent a distinct, novel group in the order *Vinavirales*, probably to form at least three new families. Using sensitive methods for sequence analysis and protein structure prediction, we describe an expanded, conserved gene core of *Vinavirales* that consists of 12 genes, including one encoding a previously undetected lysin protein.

## 2. Methods

### 2.1. Screening Gut Metagenomes for DJR MCP

Of the 33,888 human gut metagenome assemblies in the European Nucleotide Archive as of March 14, 2021 (https://www.ebi.ac.uk/ena; taxon ID 408170 “human gut metagenome”, analysis type “sequence assembly”), 23,119 assemblies available for download were retrieved. Protein-coding genes were predicted using Prodigal, v. 2.6.3, Hyatt et al., Oak Ridge, USA, in the metagenomic mode [25]. A set of HMM profiles was created using hhmake [26] from curated multiple sequence alignments of DJR MCP of Tectiviridae [8], Nucleocytovirecetes [27,28], and virophages [29]. The predicted proteins were searched against the set of DJR MCP profiles using HHSEARCH [26], with the e-value cutoff of 0.05 (see Appendix A for profile descriptions and HHSEARCH results). The search was followed with manual curation of lower-scoring candidates (true positives were recognized by the pattern of conserved residues, compatible with known DJR MCPs, and by the absence of other conserved domains recognizable with the NCBI conserved domain database, https://www.ncbi.nlm.nih.gov/Structure/cdd/wrpsb.cgi), which resulted in the identification of 226 DJR MCPs encoded in 218 contigs (Appendix A). The initial DJR profiles, nucleotide sequences and predicted proteins from the detected DJR MCP-encoding gut contigs are available at https://ftp.ncbi.nlm.nih.gov/pub/yutinn/Vinavirales_2022.

### 2.2. Classification of DJR MCP-Encoding Gut Metagenome Contigs

Homologs of human gut DJR MCPs were identified by running BLASTP against the NCBI non-redundant protein sequence database (NR) [30] with an e-value threshold of 10^−6^. The entire set of the DJR MCP sequences was clustered using MMSEQS2 [31], with a similarity threshold 0.5; the resulting protein clusters were aligned using MUSCLE5 [32] and iteratively compared to each other using HHSEARCH to establish the relationships among the clusters. Groups of related sequences were aligned using MUSCLE5 [32]; alignments were compared to the rest of the set, again using HHSEARCH. The approximate maximum likelihood phylogenetic tree was reconstructed using FastTree with a WAG evolutionary model and Gamma-distributed site rates [33] (Appendix A).

### 2.3. Abundance Calculation for the Major Capsid Protein Gene

To estimate the abundance of HK97 and DJR MCP genes in human metagenomes, 3995 datasets of Illumina human gut metagenome reads, available at the Sequence Read Archive (https://www.ncbi.nlm.nih.gov/sra) as of December 10, 2021, were used. The reads from each dataset were mapped with Kraken2 [34] to the HK97 MCP genes [24] and to the DJR MCP genes identified in this work (Appendix A).

### 2.4. Host Assignment for Members of Vinavirales from the Human Gut

Two approaches were used to infer the hosts for members of *Vinavirales* from the human gut. The first approach leverages the fact that most of these viruses are either found as prophages in the human gut metagenome or that related viruses have been sequenced as prophages along with their host genomes. For all DJR MCP-containing contigs, all predicted proteins were searched using BLASTP against a database consisting of proteins encoded in 24,757 high-quality complete genome assemblies from RefSeq and GenBank (https://ftp.ncbi.nlm.nih.gov/genomes/ASSEMBLY_REPORTS/), downloaded in November 20, 2021. Best hits for each protein from each MCP-encoding contig were recorded; the lowest-level taxon that accounted for at least half of the hits was taken as the likely source taxon for the prophage (Appendix A).

The second approach relies on matching CRISPR spacers to viral sequences. First, CRISPR arrays were predicted in all assembled human gut metagenome contigs over 10 kbp long, using the Minced v. 0.4.2 tool, Bland et al., Jackson, USA [35]. The entire set of predicted spacers was compared to putative phage sequences (DJR MCP containing gut metagenome contigs, ±8 kbp around the MCP), using BLASTN (blastn-short method; e-value threshold of 10^−15^; identity threshold of 95%). The taxonomy of each CRISPR-containing contig that produced a match into *Vinavirales* was assigned using *k*-mer content analysis with Kraken2 [34] (Appendix A).

### 2.5. Phylogenetic Analysis of Vinavirales MCP

Using *Vinavirales* MCPs detected in human gut metagenomes as queries, homologous protein sequences were extracted from GenBank; previously detected PM2 group MCPs [8] were added to the sequence set. The protein sequences were clustered at 80% identity and aligned using MUSCLE5; a phylogenetic tree was constructed from this alignment using IQ-TREE (LG + F + R5 model chosen according to BIC by built-in model finder) [36]. The alignment, the tree, and the IQ-TREE output are available at https://ftp.ncbi.nlm.nih.gov/pub/yutinn/Vinavirales_2022.

### 2.6. Gene Composition and Protein Function Prediction for Members of Vinavirales

A set of 54 (pro)phage genome sequences was selected to represent the MCP sequence diversity with the preference for known phages and contigs that included extended DNA sequences upstream and downstream of the MCP gene. For the purpose of genome annotation, ORFs were predicted in contigs using Prodigal in the metagenomic mode. Amino acid sequences were initially clustered using MMSEQS2 with the similarity threshold 0.5; the resulting protein clusters were aligned using MUSCLE5 and iteratively compared to each other using HHSEARCH. Highly similar clusters (alignment footprint coverage threshold 0.5; relative sequence similarity threshold 0.05) were progressively aligned to each other using HHALIGN. Cluster alignments were compared to publicly available profile databases (DB_mmCIF70_21_Mar, Pfam-A_v35, Uniprot-SwissProt-viral70_3_Nov_2021, and NCBI_Conserved_Domains (CD)_v3.18) using HHPRED. Protein annotations, cluster assignments and cluster annotations are in Appendix A. Conserved protein profiles are available at https://ftp.ncbi.nlm.nih.gov/pub/yutinn/Vinavirales_2022/vinavirales_conserved_proteins.

### 2.7. Prophage Boundaries and Prophage vs. Phage Calculations for Human Gut Vinavirales

Genome fragments containing ±15 kbp around the MCP of the 176 *Vinavirales* contigs were extracted and translated using Prodigal in the metagenomic mode. Predicted proteins were annotated using PSI-BLAST with profiles from NCBI CDD, PFAM and conserved *Vinavirales* profiles identified in this study. Phage boundaries were identified as corresponding to P16 and P15 homologs (Appendix A). The contig was classified as a prophage if it contained at least one gene outside of the phage boundaries and as a phage otherwise. The contigs were classified as complete if both boundary genes (P16 and P15) were present and as partial otherwise.

### 2.8. Protein Structure Prediction with Alphafold2 and Protein Structure Comparison

In order to predict protein structures from the alignments, a local version of ColabFold v. 1.2 [37] (Mirdita et al., Cambridge, USA; alphafold2_batch) was downloaded on 15 December 2021. ColabFold was run with default settings with the following modifications: input was a precomputed alignment generated as described above, and --num-model 1 and –num-recycle 3 were set to 1 and 3, in order to minimize the computational cost.

Structures retrieved with ColabFold were compared against reference protein structure databases using DALI [38] and foldseek [39]. For DALI, a mirror of the pdb database as of December 2021 was dereplicated at 70% identity and transformed into a DALI compatible database. Apart from using the 70% dereplicated database, DALI was run with default parameters. For foldseek, pre-assembled databases (Alphafold/Proteome, Alphafold/Swiss-Prot and PDB) were fetched in January 2022 from foldseek. Predicted structures were compared against all three foldseek databases via ‘foldseek easy-search input.pdb db out.m8 tmpFolder’.

### 2.9. Protein Structure Visualization

Protein structures were visualized with UCSF ChimeraX [40]. Superposition of the predicted structure for P19 and endolysin pdb 7q47 chain A [41] was obtained using the ChimeraX matchmaker tool with 7q47 chain A as the reference structure and the ‘Best-aligning pair of chains between reference and match structure’ option.

## 3. Results

### 3.1. DJR MCP-Encoding Contigs in Gut Metagenomes

We screened 23,119 EMBL-assembled human gut metagenomes for DJR MCP by searching the set of open reading frames (ORFs) from metagenome contigs using HHSEARCH, with a collection of DJR MCP profiles representing different groups of viruses within *Varidnaviria* used as queries (see Methods). This search and subsequent manual curation of lower-scoring candidates yielded 226 DJR MCP sequences encoded in 218 contigs (Appendix A). This set of DJR MCPs was augmented with closely related homologs, that were retrieved by running BLASTP against the NCBI non-redundant protein sequence database and clustered using MMSEQS2. The resulting protein clusters were aligned and compared to each other using HHSEARCH.

This analysis revealed five distinct clusters of DJR MCPs (Figure 1; Appendix A). The largest of these clusters included 176 of the 218 (80%) contigs that encoded DJR MCPs related to those of phages in the order *Vinavirales*. These sequences belong to and expand the previously described PM2-like phage group [8] and are discussed in detail below. The second largest cluster, 22 proteins on 16 contigs, grouped with the MCPs of large and giant mimi-, phycodna- and iridoviruses from the phylum *Nucleocytoviricota* [42]. All these viruses infect unicellular eukaryotes that are not considered to be native human gut inhabitants and, accordingly, neither are their viruses. Twelve MCPs on 10 contigs showed (in some cases, distant) similarity to the MCPs of virophages, which parasitize giant viruses and, like the latter, are also unlikely to be native members of the human gut virome. Ten closely similar MCPs on 10 contigs belong to fish polintoviruses; these sequences group with fish, reptile, and four ‘adintovirus’ MCP sequences from GenBank, so their native association with human gut commensals is questionable, at best. Finally, six MCPs on six contigs belong to the ‘Bam35 group’ of tectiviruses [8,43]. As well as the MCP, these contigs encode the FtsK-like packaging ATPase, peptidase M23, family B DNA polymerase and a lysozyme; their closest relatives among identified phages are *Bacillus* tectiviruses Wip1, AP50, and V Ba and the *Thermus* phage phiKo [44] (AYJ74688; Appendix A).

The abundance of members of *Duplodnaviria* and *Varidnaviria* in the gut metagenomes differed about 1500 fold (assessed by the counts of reads mapped to the HK97 and DJR MCP genes, respectively; Figure 1B). DJR MCP genes were detected in 7% of the datasets (294 of the 3995), as compared to the 96% of the datasets (3851 out of 3995) in which HK97 MCP genes were detected. This stark contrast reaffirms that members of *Varidnaviria* are a minor component of the human gut virome. While members of the order *Crassvirales*, the most abundant phages of the realm *Duplodnaviria* in the human gut, accounted for 28% of the HK97 MCP reads (6,346,752 of the 22,722,487), the *Vinavirales*-like group of DJR MCPs dominated the varidnavirus fraction of the gut virome in terms of both contig number and read counts (Figure 1A). Indeed, 92% of the reads that mapped to DJR MCP-encoding contigs belonged to this group (13,832 of 15101).

Of the 176 contigs that represent *Vinavirales* in the analyzed human gut metagenomes, 142 were identified as prophages, as attested by the presence of host flanking genes (see Section 2), and four probably originated from virus particles, whereas the remaining 30 could not be identified as either phage or prophage. Thus, phages of the order *Vinavirales* are mostly present in the gut as lysogens, with apparently few lytic infections.

The presence of bacterial genes flanking prophages allowed host assignment for the majority of the Vinavirales members identified in the gut metagenomes. Additionally, we sought to identify hosts by analysis of CRISPR spacer matches to the (pro)phage sequences. To this end, CRISPR arrays were predicted in all assembled human gut metagenome contigs and the spacer sequences were compared to the (pro)phage sequences (see Methods). Altogether, we identified 115 CRISPR spacers, matching 102 of the 176 *Vinavirales* contigs, and providing for host assignment for the majority of the (pro)phages (Appendix A).

### 3.2. Phylogenomics of Vinavirales

Using MCPs of the viruses of the order *Vinavirales* detected in human gut metagenomes as queries, we extracted related GenBank protein sequences; the previously detected PM2 group MCPs [8] were also added to the sequence set. The protein sequences were clustered at 80% identity, aligned, and used to construct a maximum likelihood phylogenetic tree (see Section 2). At coarse grain, the *Vinavirales* MCPs formed five strongly supported major clades (A to E). Clades B to E were largely consistent in terms of the taxonomic affiliation of the inferred hosts, whereas clade A was heterogeneous (Figure 2).

The largest clade, A, is composed of gut metagenome sequences and contigs from various environments and includes no cultured phages. Viruses in this clade appear to be associated mostly with alphaproteobacterial hosts, along with some that are likely associated with Verrucomicrobia. The gut metagenome sequences all belonged to this clade and formed three distinct groups, which we denoted HGV-1,2,3 (HGV, human gut *Vinavirales*). The HGV-1 group is by far the most abundant among gut varidnaviruses in terms of read counts, accounting for 95% (13,147 of 15,101) of the *Vinavirales*-specific reads and 87% (13,147 of 13,832) of all the reads mapping to DJR-encoding contigs in the human gut metagenomes (Appendix A).

Clades B and C include all known *Corticoviridae* and *Autolykiviridae*, respectively, both of which infect Gammaproteobacteria. The remaining two clades consist of prophages from mainly Betaproteobacteria (clade D) and Gammaproteobacteria (clade E), without any known phages or gut metagenome sequences.

Genome organization and gene content of (pro)phages of clades A, B, and C, which include either HGV or established *Vinavirales* families, *Corticoviridae* and *Autolykiviridae*, were analyzed in detail for a set of 54 genomes selected to represent the MCP sequence diversity, with the preference for contigs with terminal repeats and contigs containing at least 15 kbp upstream and downstream of the MCP gene. All predicted proteins encoded in these genomes were searched for known domains and distant homologs (Appendix A; see Section 2).

Figure 3 shows the genome organizations of representative (pro)phages of *Vinavirales* clades A, B, and C. Three protein families were represented in each (pro)phage genome, namely, MCP, packaging ATPase and a protein homologous to P1 of *Pseudoalteromonas* phage PM2, a multidomain spike protein containing the penton domain [45]. Upstream of the ATPase, nearly all genomes encode an uncharacterized protein homologous to ORF h of PM2. Sequence similarity among these proteins is low, so that homology could be demonstrated only through a customized HHpred search. To this end, protein sequences were grouped into five clusters and a multiple protein sequence alignment was constructed for each cluster (dubbed cons_a1–a5) and used as queries for HHpred search. Three of the five alignments (cons_a1, a2, a4) showed similarity to either the Gp-h protein of phage PM2 or P10 protein of *Enterobacteria* phage PRD1 (HHPred probability 99%, 25%, and 95%, respectively), whereas one (cons_a3) showed similarity to both the Gp-h and P10 proteins (probability 97% and 96%, respectively). Protein P10 of phage PRD1 is an assembly factor involved in the formation of the membrane vesicle that in mature virions encloses the viral genome and is enclosed within the icosahedral protein capsid [46]. Our present findings establish homology between PM2 Gp-h and PRD1 P10, further extending the similarities between virion assembly pathways in the two families, and show that Gp-h-like proteins are conserved across the *Vinavirales*.

Between the MCP and spike genes, most of the genomes encode predicted membrane proteins homologous to PM2 proteins P3, P8, and P10, as well as a GNAT family acetyltransferase (missing in PM2, but present in *Pseudoalteromonas* phage GX1010, *Vibrio* phage fNo16, and cultured members of *Autolykiviridae*). Some of the structural proteins encoded in this region probably remain unannotated in some (pro)phage sequences because of the low sequence conservation and/or because they are missed by the gene calling software due to their small size.

In many genomes, there is a gene between ATPase and MCP encoding a small (40–50 aa) predicted membrane protein. In this case, annotation is obviously hampered by the short protein length, but some of these proteins were readily alignable with P7 of PM2, a minor structural protein located in the viral membrane [47] that belongs, along with MCP and the ATPase, to the “Corticoviridae self genes” [13]. The present observations indicate that this gene is apparently present across *Vinavirales*, not only in *Corticoviridae*.

In PM2 phage, P17 and P18 proteins are lysis factors; P17 represents a distinct class of holins involved in depolarization and permeabilization of the cytoplasmic membrane, whereas P18 is required for the disruption of the bacterial outer membrane [48]. However, no peptidoglycan-digesting enzyme has been detected in PM2, and therefore it has been proposed that, unusually for a phage, PM2 relies on a host lytic enzyme. We found that a protein homologous to the PM2 structural protein P5 is encoded by all clade B phages and most of the clade C phages (denoted ‘peptydoglycan hydrolase’ in Autolykiviridae [3]). Profiles constructed from multiple sequence alignments of these proteins showed significant similarity to LytD superfamily lysins [49] in HHpred searches (Appendix A). AlphaFold2 structure prediction for the multiple sequence alignment of P5 protein and its homologs and superposition with the experimentally solved structure of phage Enc34 endolysin (pdb 7q47), the closest structure identified by a Dali search, supports the annotation of P5 protein and its homologs as endolysins (Figure 4; https://ftp.ncbi.nlm.nih.gov/pub/yutinn/Vinavirales_2022/P5_lysin_structure/). In addition to the overall fold, the proposed catalytic triad (Trp93, Tyr105 and Glu137) within the substrate binding groove found in homologous structures is conserved (Figure 4C, [41]). Thus, PM2 and other phages in clades C and B of *Vinavirales* all encode a pepdtidoglycan-hydrolyzing enzyme (Figure 3). Notably, clade A genomes lack a P5 homolog; instead, in these phages, the gene located in the genome region corresponding to the P5 genes of PM2 encodes an unrelated cell wall hydrolase of the SleB superfamily [50] (denoted CWH on Figure 3). This difference likely reflects the distinct host ranges of phages from clade A and clades B and C (Alpha- and Gammaproteobacteria, respectively). Four genomes in clade A and one genome in clade C encode lytic enzymes unrelated to these two, namely, four lysins of the MltE superfamily (JAACAO010000289, JAESRS010000067, NZ_RBIG01000001, DRNT01000151) and one (CEPX01414995) of the T4lyz-like superfamily (denoted ‘other_CWH’ on Figure 3). In the genomes of phages predicted to infect Verrucomicrobia, no homologs of any of the known cell wall degrading enzymes were detected, suggesting that these phages might encode an unknown lysin; however, it cannot be ruled out that these are inactivated prophages, because they also lack conserved genes encoding transcriptional regulators P15 and P16 [51]. On the other hand, it has been shown using transposon insertion mutagenesis that P15 is one of the few PM2 genes dispensable for infectivity [52].

Lysis factors that were homologous or possibly functionally analogous to P17 and P18 of PM2 phage were predicted in many *Vinavirales* genomes, among the products of the corresponding genes occupying the equivalent genome region (Figure 3). The homology among these proteins could not be easily established, due to their small size and low sequence similarity.

Transcription regulators orthologous to P16 of PM2 phage are conserved in most phages of clades A, B, and C, and orthologs of transcription regulator P14 are represented in most of clade B and some of the genomes in clades C and A. Orthologs of transcription regulator P15 of PM2 are conserved across clades B and C, but completely absent in clade A. However, members of clade A encode another, distinct predicted transcription factor encoded in the same genome locus as P15 (denoted TR_Z1 in Figure 3). The presence orabsence of the mutually exclusive P15 and TR_Z1 corellates with predicted host ranges. Verrucomicrobia-infecting phages in clade A lack a TR_Z1 homolog. Some phage genomes encode additional predicted transcription regulators containing the HTH domain.

In previous analyses, only two proteins shared between *Autolykiviridae* and *Corticoviridae* have been identified, MCP and ATPase [3]. Here we show that these two phage families have 15 genes in common. The additional conserved genes encode spike protein; structural proteins P3, P8, and P10; GNAT acetyltransferase; endolysin; uncharacterized ORF-h homolog; structural potein homologous to P7; lysis factors homologous or analogous to P17 and P18; and transcriptional regulators homologous to P14, P15 and P16.

Protein profiles developed in the analysis of clades A, B and C, were used to annotate contigs of clades D and E (Figure 2). These viruses are associated with either Betaproteobacteria (most of clade D) or Gammaproteobacteria (the entire clade E and part of clade D). The clade E and D contigs were subjected to the same protein prediction and annotation procedure as those of clades A, B and C. Most of the conserved proteins identified in the first round of the analysis were also detected in clade E and D contigs, with the exception of the clade A-specific CWH and TR_Z1 (Appendix A).

To define the core gene set of *Vinavirales*, for each conserved protein family and for each clade, we calculated the fraction of genomes where the given gene was detected (Verrucomicrobium prophages, the top eight genomes in Figure 3, were excluded as they were extremely divergent, possibly, degenerate ones). Various replication proteins (see below) and different endolysins were each counted as one protein family. The core genes were defined as those represented in more than 50% of the members of at least three of the five clades. This resulted in a *Vinavirales* core of 12 genes (Figure 5A). Several proteins are highly conserved in individual clades but failed to make the core, including P15_TR, which is present in most members of clade B (83%), sparsely present in clades C,D,E (50% or less), and replaced with TR_Z1 in clade A., consistent with the dispensability of P15 gene in phage PM2 [52]. Lytic factors P17, P18, and various holins, although universally present, were not included in the core set because these proteins are short and highly divergent, which hampers homology detection. The small structural protein P7, probably encoded between the ATPase and MCP genes, failed to make the core set for the same reason: it is likely ancestral (sparsely present in all five clades), but is too short to establish homology.

Notably, phages within *Vinavirales* that are predicted to infect bacteria of different taxa typically encode distinct lytic enzymes. Thus, phages of Alphaproteobacteria encode a cell wall hydrolase of the SleB superfamily; phages of Gammaproteobacteria, despite belonging to different clades, all encode the PM2 P5 homolog of LytD superfamily, whereas phages predicted to infect Betaproteobacteria encode a T4lyz-like protein (Figure 5B).

Although the great majority of the members of the *Vinavirales* detected in human gut metagenomes appear to be prophages, only HGV-2 contigs encode a homolog of the DDE transposases of the IS630 family (Figure 3; Appendix A), which likely mediate the prophage integration. The integration mechanisms for other *Vinavirales* prophages remain unclear.

## 4. Discussion

The human gut virome is probably one of the most thoroughly studied viromes on earth. However, most of the metagenomic analyses of viruses in the human gut have focused on the dominant group, the tailed viruses of the class *Caudoviricetes* (realm *Duplodnaviria*), whereas other viruses have largely been neglected. Here, we searched gut metagenomes for members of the second major realm of dsDNA viruses, *Varidnaviria*, by applying sensitive methods to detect genes encoding DJR MCP. The results of this search indicate that members of *Varidnaviria* are a minor component of the human gut virome, in the order of 1%. Furthermore, a substantial majority of these viruses are found in the form of prophages, suggesting that their activity in the gut is low, with few lytic infections ongoing at any given time. These observations sharply contrast the results of metagenomics analysis of marine habitats, where varidnaviruses appear to represent the majority [2,3]. The causes of such contrasting virome compositions remain unclear but, at least in part, are likely to reflect the differences in abundance of different bacterial taxa in the gut compared to the oceans, and distinct, even if overlapping, host ranges of bacterial duplodnaviruses and varidnaviruses.

Despite their comparative scarcity, phylogenomic analysis of varidnaviruses from the gut revealed notable trends. Apart from eukaryotic giant viruses and virophages that most likely come from food-borne contamination, the great majority of gut varidnaviruses belong to the order *Vinavirales*. So far, this order has included two virus families, *Corticoviridae* and *Autolykiviridae*, but the present analysis of gut metagenomes suggests the existence of at least three new families (clades A, D and E in the phylogenetic tree of *Vinaviriales*; Figure 2). We propose to name these families after the wife of Autolykus, the namesake of *Autolykiviridae*. In keeping with Autolykus’ (and *Vinavirales’*) elusive nature, she was reported by different sources using three different names, Mestra, Neaera and Amphithea, hence the proposed family names Mestraviridae (clade A), Neaeraviridae (Clade D) and Amphitheaviridae (clade E). This expansion of the order *Vinaviriales,* combined with using sensitive methods for sequence comparison and protein structure modeling, has enabled a substantial extension of the core of conserved genes for this virus order, from only two to 12 genes. For several previously uncharacterized genes, functional prediction has become possible. In particular, lytic enzymes were predicted in the model bacteriophage PM2 and its relatives, which have previously been thought to lack such proteins [13,48].

The prophage state of most of the members of *Vinavirales* facilitates assignment of bacterial hosts and, combined with the analysis of CRISPR spacer matches, allowed us to predict the hosts for most of the identified prophages and phages. Mostly, the members of the order *Vinavirales* in the human gut virome were predicted to infect Proteobacteria, and in particular Alphaproteobacteria, to which the most abundant prophages were assigned. The host distribution, at least, roughly corresponds to the major clades in the phylogenetic tree of *Vinavirales*, suggesting a degree of coevolution between bacteria and phages. Furthermore, we found that phages infecting Alpha-, Beta-, and Gammaproteobacteria encode distinct lytic enzymes, which suggests that differences in the host cell wall structure limit the host range of the phages and determine the optimal choice of the lysin recruited by the respective phages. Another phenomenon with potentially interesting biological implications is the frequent replacement of replication genes, even among otherwise closely related (pro)phages, supporting the previous conclusion from the analysis of aquatic PM2-like prophages [13]. This finding recapitulates a similar observation for crAss-like phages [20], suggesting the existence of antiphage mechanisms targeting phage replication machineries that might be widespread in bacteria but remain to be identified. Alternatively, the replication module replacement might be promoted during coinfections, due a phenomenon analogous to plasmid incompatibility, whereby closely related replicons cannot stably coexist in the same cell [53].

## 5. Conclusions

The results of this analysis of the human gut metagenomes shows that, in a sharp contrast to marine viromes, tailless bacteriophages of the realm *Varidnaviria* represent a minor component of the gut virome. Furthermore, these viruses are primarily found in the form of prophages, suggesting that few lytic infections are occurring in the gut at any given time. Most of the varidnavirus prophages in the gut belong to the order *Vinavirales*, and phylogenetic analysis of the MCP suggests that at least three new families should be established within *Vinavirales* to accommodate the diversity of these prophages. Our comparative genome analysis expands the core gene set of *Vinavirales* to 12 genes that are shared by most of these viruses, including genes encoding previously undetected lysis enzymes. We demonstrate that replication system components are frequently replaced in the genomes of *Vinavirales*, suggestive of selective pressure for escape from yet unknown host defenses or avoidance of incompatibility with coinfecting related viruses. These findings extend the knowledge of the human gut virome to a group of viruses that has been virtually overlooked in previous work. Together with the results of previous metagenomics studies, the findings reported here show that major expansion of different groups of viruses provided by metagenomics not only reveals the composition of viromes, but also advances phylogenomics and enables substantial improvements in the functional annotation of virus genomes.

## Figures and Tables

**Figure 1 viruses-14-01842-f001:**
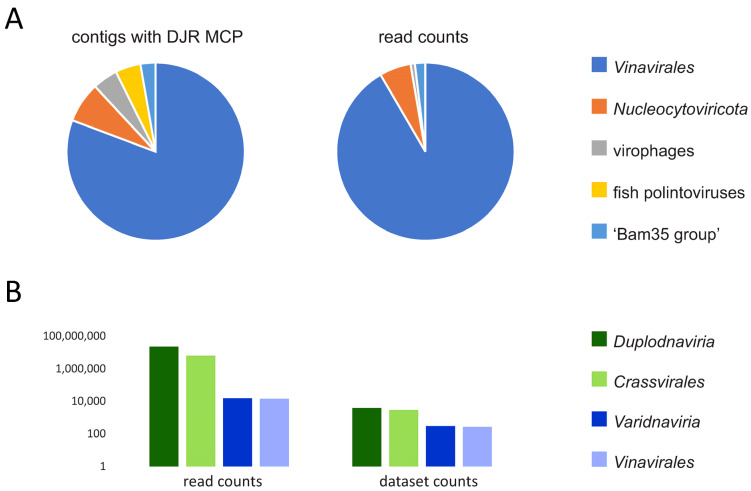
Abundance of DJR MCP in human gut metagenomes. (**A**) Number of contigs assigned to different groups of varidnaviruses (**left**) and number of reads mapped to DJR MCPs of each of these groups (**right**). (**B**) Comparison of the number of the counts of reads and datasets (metagenomes) for HK97 MCP (*Duplodnaviria*) and DJR MCP (*Varidnaviria*).

**Figure 2 viruses-14-01842-f002:**
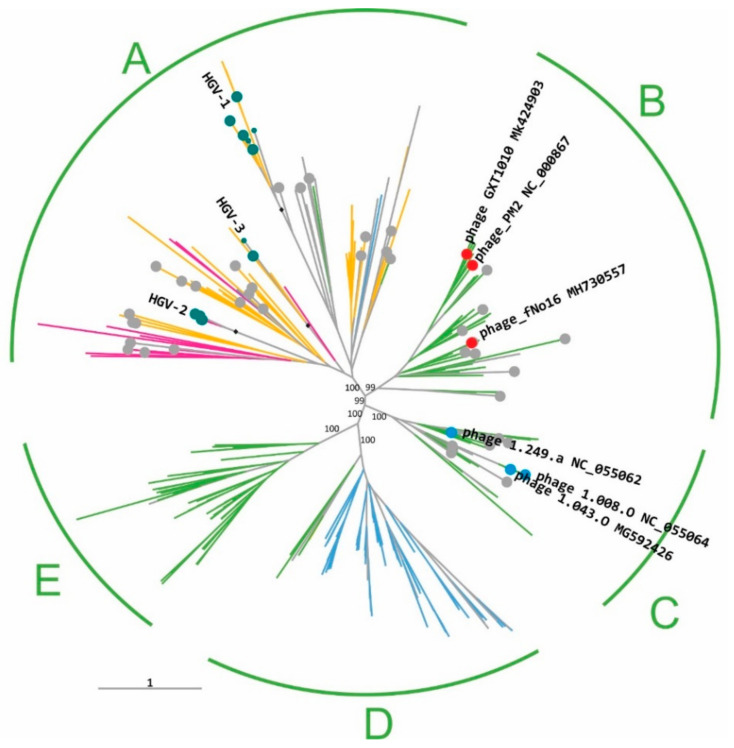
Phylogenetic tree of the MCP of viruses of the orfer *Vinavirales*. Branch color indicates predicted hosts: green, gammaproteobacteria; orange, alphaproteobacteria; blue, betaproteobacteria; magenta, Verrucomicrobia. Human gut sequences, cultured Corticoviridae and Autolykiviridae are labeled (teal, red, and blue circles, respectively). Circles mark 54 representative genomes/contigs selected for further analysis. The 5 major clades are denoted A–E, and the bootstrap support values are indicated for each of these clades. The 3 groups of MCPs from human gut microbiomes are denoted HGV1-3. The branches leading to the groups HGV1-3 all had bootstrap support values of 100 and are marked with black diamonds.

**Figure 3 viruses-14-01842-f003:**
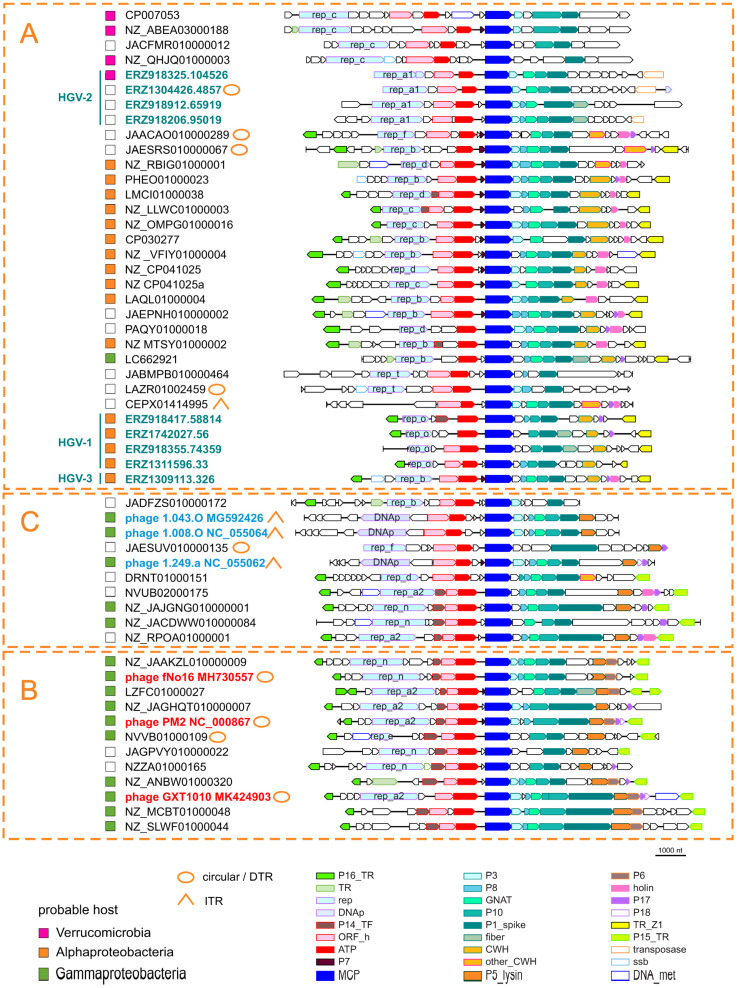
Genome organization of *Vinavirales*. Genome maps of the clades **A**, **B**, **C** representatives marked on the Figure 2 tree are shown. Font color indicates the following: teal, human gut metagenome contigs identified in the current study; blue, cultured Autolykiviridae; red, cultured Corticoviridae. Homologous genes are denoted by colored block arrows; the color code is given in the bottom. Uncharacterized genes are shown by uncolored arrows. Abbreviations: DTR, direct terminal repeats; ITR, inverted terminal repeats; P16_TR, transcriptional regulator P16 of PM2 phage; TR, transcriptional regulator; rep, replication protein; DNAp, family B DNA polymerase; P14_TF, transcriptional factor P14 of PM2 phage; ORF_h, putative assembly factor homologous to ORF h of PM2 and P10 of phage PRD1; ATP, packaging ATPase; P7, minor structurel protein P7 of PM2 phage; MCP, major capsid protein; P3, membrane protein P3 of PM2 phage; P8, membrane protein P8 of PM2 phage; GNAT, GNAT family acetyltransferase; P10, membrane protein P10 of PM2 phage; P1_spike, protein homologous to P1 of PM2 phage; fiber, tail fiber protein homolog; CWH, SleB superfamily cell wall hydrolase; other_CWH, cell wall hydrolase; P5_lysin, lytic enzyme P5 of PM2 phage; P6, major membrane protein P6 of PM2 phage; P17, lysis factors P17 of PM2 phage; P18, lysis factors P18 of PM2 phage; TR_Z1, transcription regulator conserved in clade A; P15_TR, transcription regulator P15 of PM2 phage; ssb, single strand DNA-binding protein; DNA_met, DNA methyltransferase.

**Figure 4 viruses-14-01842-f004:**
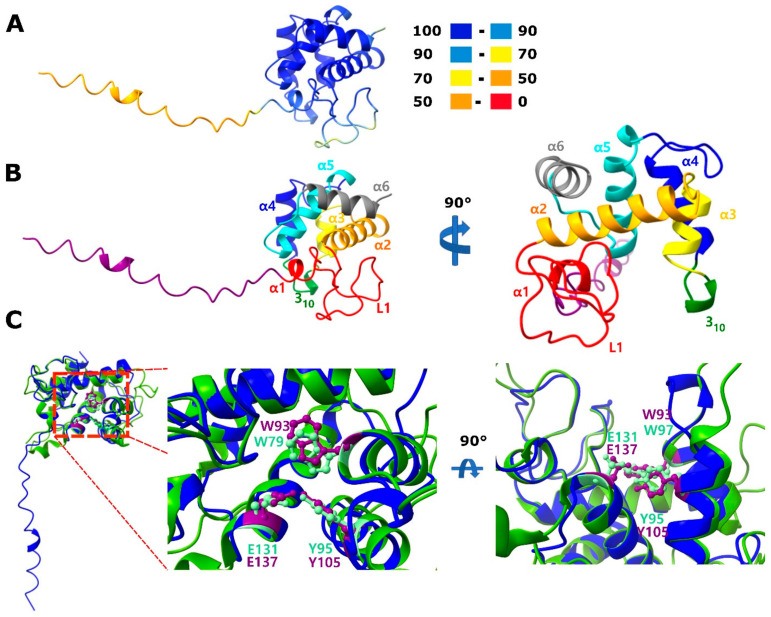
Predicted structure of Pseudoalteromonas phage PM2 protein P5, a putative endolysin. (**A**) AlphaFold2 structure prediction for P5 homologs. Coloring is according to the b-factor as reported by Alphafold2: 100–90: extremely likely, 90–70: well supported, 70–50: low confidence, 50–0: probably disordered. (**B**) Coloring and nomenclature of helices and loops for the portion of P5 predicted with high confidence. Loop 1 (L1), α2, the single one-turn helix 310 and the N-terminal portion of α3 form the catalytic groove as suggested for a recently described endolysin [41]. Coloring from N-to C-terminus: purple, red, orange, yellow, green, blue, turquoise and grey. (**C**) Overlay of the predicted P5 structure shown in (**A**,**B**) (blue) over the experimentally resolved catalytic domain of phage Enc34 endolysin (pdb 7q47, chain A, green, [41]. Conserved amino acid residues hypothesized to be involved in the catalytic activity are highlighted for the reference (light green, W79, Y95 and E131) and P5 (purple, W93, Y105 and E137). Structure figures were prepared with ChimeraX. Amino acid positions are indicated for the catalytic domain as described in 7q47, not for the full protein. In P5 (NP_049909), one of the homologues included in the alignment for the alphafold2 structure prediction, these sites correspond to W96, Y107 and E139.

**Figure 5 viruses-14-01842-f005:**
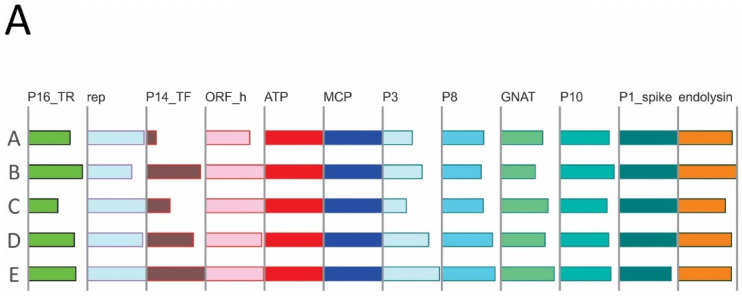
Core genes of *Vinavirales*. (**A**) The 12 genes defined as *Vinavirales* core by their presence in all 5 clades (A to E). The bar lengths correspond to the fraction of the genomes in a clade carrying the respective gene. (**B**) Endolysin repertoire of *Vinavirales* largely follows the predicted host taxonomy. Various subsets of replication proteins, such as DNA polymerase B, non-homologous rolling circle replication initiation endonucleases of the Rep_trans (PF02486) and HUH superfamilies, DnaD-like winged HTH domain replication initiators, and others (Appendix A), were identified in nearly all members of *Vinavirales*. These genes are apparently subject to frequent non-orthologous displacement during the evolution of this phage group, similarly to the replication proteins of crAss-like tailed phages [20]. An example of replication protein replacement is shown in Figure 6. The *Paremcibacter congregatus* strain ZYLT genome encodes two prophages that belong to clade A. Strikingly, these prophages are nearly identical for most of their length, with the notable exception of the replication genes (divergent Rep_1 family rolling circle replication endonucleases of the HUH superfamily; pfam01446) and two or three ORFs upstream of these genes. The replication proteins and the upstream ORFs lack detectble sequence similarity between the two prophages, and the upstream ORFs in each phage encode short (80–110 aa) proteins with no detectable homologs in current sequence databases.

**Figure 6 viruses-14-01842-f006:**
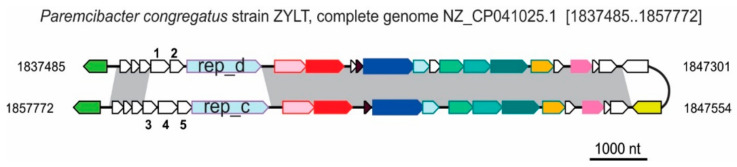
Replacement of a replication protein gene in closely related prophages. Two closely related prophages from the *Paremcibacter congregatus* genome are aligned; regions with more than 90% identical nucleotides are connected by gray shaded shapes. Genes are colored as on Figure 3. Five numbered genes are: 1, NZ_CP041025_15; 2, NZ_CP041025_16; 3, NZ_CP041025_51; 4, NZ_CP041025_50; 5, NZ_CP041025_49 (Appendix A).

## Data Availability

DJR MCP profiles used for gut metagenome screening for *Varidnaviria,* nucleotide sequences of all *Varidnaviria* contigs identified in gut metagenomes, multiple alignments of conserved Vinavirales proteins identified in this study, protein sequences and annotations of representative Vinavirales genomes are available at https://ftp.ncbi.nih.gov/pub/yutinn/Vinavirales_2022/.

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
