# Peer review of "Varidnaviruses in the Human Gut: A Major Expansion of the Order Vinavirales"

_viruses, 2022, doi:10.3390/v14091842_

Round 1

Reviewer 1 Report

Yutin et al. have extensively analysed the realm of Varidnaviria, a group of viruses mainly overlooked in the gut, using metagenomics data. They shed some light on the composition go these viruses in the human gut and provided some information of their genomic structure and function. However, the manuscript is well written, and the study is solid, there are a few concerns, mainly about the interpretation of these findings: 

  1. As the authors have looked into metagenomes and not VLP extracted data as far as I can say, I believe their conclusion over these viruses being mainly prophages is invalid. Otherwise, this needs to be clarified in the text. 
  2. However, I understand the author's strategy for differentiating proviruses from non-integrated viruses. I was wondering if they looked into lysogeny-relevant genes in these viruses? Or if they tried to predict these genes in case, they are largely unknown? 
  3. How about their geographical distribution? Do they know if these viruses are widespread or specific to a geographical region?
  4. Line 447, I disagree with the authors that gut virome is studied most and suggest they change this sentence to "is probably one of the most…"
  5. Line 455, how did the authors determine the activity of these phages in the gut? and how they define active phages vs inactive phages? I think this needs some clarification. Tools like PropagAtE (doi: 10.1128/msystems.00084-22) might be useful here. 

Author Response

Yutin et al. have extensively analysed the realm of Varidnaviria, a group of viruses mainly overlooked in the gut, using metagenomics data. They shed some light on the composition go these viruses in the human gut and provided some information of their genomic structure and function. However, the manuscript is well written, and the study is solid, there are a few concerns, mainly about the interpretation of these findings: 

  1. As the authors have looked into metagenomes and not VLP extracted data as far as I can say, I believe their conclusion over these viruses being mainly prophages is invalid. Otherwise, this needs to be clarified in the text. 

The contigs were classified as prophages on the basis of the presence of host genes flanking the phage genome as indicated in the text (both Methods and Results). To avoid any ambiguity, in the revision, we also mention this in the Abstract.

  1. However, I understand the author's strategy for differentiating proviruses from non-integrated viruses. I was wondering if they looked into lysogeny-relevant genes in these viruses? Or if they tried to predict these genes in case, they are largely unknown? 

We appreciate this highly relevant comment. We included the following in the revised manuscript “Although the great majority of the members of the Vinavirales detected in the human gut metagenomes appear to be prophages, only HGV-2 contigs encode a homolog of the DDE transposases of the IS630 family (Figure 3; Supplementary file 4) which likely mediate the prophage integration. The integration mechanisms for other Vinavirales prophages remains unclear.“

  1. How about their geographical distribution? Do they know if these viruses are widespread or specific to a geographical region?

We did not immediately detect any strong bias, the scattered geographically. We did not think this was important to point out in the manuscript given that Vinavirales are found only in a minority of gut microbiomes.

  1. Line 447, I disagree with the authors that gut virome is studied most and suggest they change this sentence to "is probably one of the most…"

 Certainly, modified as suggested.

  1. Line 455, how did the authors determine the activity of these phages in the gut? and how they define active phages vs inactive phages? I think this needs some clarification. Tools like PropagAtE (doi: 10.1128/msystems.00084-22) might be useful here. 

Perhaps, using “active” was misleading here. We simply meant that the fact the substantial majority of the members of Vinavirales were prophages rather than free phages suggests there are few lytic infections. So “active” replaced with “lytic” and softened the language in the Conclusions, too. PropagAtE is an interesting but we believe somewhat insufficiently tested tool. We found this to be somewhat beyond the scope of the study.

Reviewer 2 Report

The authors have written a really good paper that provides important insights into the diversity of the human gut virome. I think it should be published in viruses. I only have some minor comments.

Line 16: repetition of prophages. I’d recommend removing one of them

Line 20-22. Not sure if I agree with the way this is presented. The author says “Previously, only the MCP and packaging ATPase genes were reported to be conserved in all members of Vinavirales.” which sets up the next sentence to perhaps reveal other genes that are present in ALL members but instead the sentence mentions that the 12 core proteins are shared by “most” of these viruses. I think it needs to be re-worded slightly.

Line 90. Extra space after Nucleocytovirecetes?

Line 94: Was there a threshold score you used to define “lower-scoring candidates”? Could you add more information on the manual curation? What were you looking for and what information informed your choices?

Lines 84-97. Do you want hhsearch to be lowercase? Is uppercase in lines 99-107. Can you be consistent.

Figure 3. Change key colors for circular /DTR and ITR to the orange in A-E. Or is there a reason they are a different color?

Line 371: Unfinished sentence “Amino acid positions are indicated according to the.” To the what?

Line 415. Misspelling of ancestral?

Line 454-455. “Furthermore, a substantial majority of these viruses are found in the form of prophages, suggesting that their activity in the gut is low, with few lytic infections ongoing at any given time.” These viruses are relatively small, are you/the field sure that the infectious particles aren’t being lost during the collection/preparation of the samples for sequencing? That might explain why they aren’t being seen. I don’t want the authors to change the language I just want to make sure other explanations have been considered. I can foresee this conclusion being used in the future so I think it is important to consider it thoroughly.

Figure 4 C. Do you ever really talk about the potential conservation of the catalytic site in the main text? I couldn’t see any mention of it. I’d suggest either adding something to main text or just remove the zoomed in panels and just show the overlay of the two, which you do mention in the text.

Linwe 583. KooninE.V. needs changing to Koonin EV

Supplementary file 1. Can the meaning of the colors be described in the file, please.

Supplementary file 2. Can you add the color key again to the final slide ‘Bam’ group, representative genome.

Supplementary file 3. Can the meaning of the colors be described in the file, please.

Supplementary file 5. Can the meaning of the colors be described in the file, please.

Supplementary file 6. Can the meaning of the colors be described in the file, please.

Supplementary file 7. Can the meaning of the colors be described in the file, please.

Supplementary file 8. Can you add the color key again.

Author Response

The authors have written a really good paper that provides important insights into the diversity of the human gut virome. I think it should be published in viruses. I only have some minor comments.

Line 16: repetition of prophages. I’d recommend removing one of them

  Corrected

Line 20-22. Not sure if I agree with the way this is presented. The author says “Previously, only the MCP and packaging ATPase genes were reported to be conserved in all members of Vinavirales.” which sets up the next sentence to perhaps reveal other genes that are present in ALL members but instead the sentence mentions that the 12 core proteins are shared by “most” of these viruses. I think it needs to be re-worded slightly.

We reworded the sentence in question as follows:

“Previously, only the MCP and packaging ATPase genes were reported as conserved core genes Vinavirales. Here we report an extended core set of 12 proteins, including MCP, packaging ATPase, and previously undetected lysis enzymes, that are shared by most of these viruses.”

Line 90. Extra space after Nucleocytovirecetes?

   Corrected

Line 94: Was there a threshold score you used to define “lower-scoring candidates”? Could you add more information on the manual curation? What were you looking for and what information informed your choices?

The sentence was expanded with “the true positives were recognized by the pattern of conserved residues, compatible with known DJR MCPs, and by the absence of other conserved domains recognizable with NCBI conserved domain database, https://www.ncbi.nlm.nih.gov/Structure/cdd/wrpsb.cgi”

Lines 84-97. Do you want hhsearch to be lowercase? Is uppercase in lines 99-107. Can you be consistent. hhsearch as a command  – small letters, as a method - HHsearch

 Corrected as suggested

Figure 3. Change key colors for circular /DTR and ITR to the orange in A-E. Or is there a reason they are a different color?

 Corrected as suggested.

Line 371: Unfinished sentence “Amino acid positions are indicated according to the.” To the what?

 We regret this sloppiness. Corrected to “Amino acid positions are indicated for the catalytic domain as described in 7q47, not for the full protein.”

Line 415. Misspelling of ancestral?

 Corrected

Line 454-455. “Furthermore, a substantial majority of these viruses are found in the form of prophages, suggesting that their activity in the gut is low, with few lytic infections ongoing at any given time.” These viruses are relatively small, are you/the field sure that the infectious particles aren’t being lost during the collection/preparation of the samples for sequencing? That might explain why they aren’t being seen. I don’t want the authors to change the language I just want to make sure other explanations have been considered. I can foresee this conclusion being used in the future so I think it is important to consider it thoroughly.

This is certainly a relevant comment. We believe we considered the issue of prophages vs free phage carefully enough, and some changes to the wording were made in response to similar comments of reviewer 1. We surely do share the reviewer’s hope that these findings will be used by others.

Figure 4 C. Do you ever really talk about the potential conservation of the catalytic site in the main text? I couldn’t see any mention of it. I’d suggest either adding something to main text or just remove the zoomed in panels and just show the overlay of the two, which you do mention in the text.

We appreciate this relevant comment. The following sentence has been added to the main text ”Besides the overall fold, the proposed catalytic triad (Trp93, Tyr105 and Glu137) within the substrate binding groove found in homologous structures is conserved (Figure 4C, [41]).”

Line 583. KooninE.V. needs changing to Koonin EV

 Corrected

Supplementary file 1. Can the meaning of the colors be described in the file, please.

The sentence “The color gradient (green to yellow) highlights the ranking of protein length and contig length values” was added to the file and to the file description in the text.

Supplementary file 2. Can you add the color key again to the final slide ‘Bam’ group, representative genome.

The descriptions of the colored genes on the “Bam’ group representative genome map are written right below the genes, so adding a separate color key legend would be abundant.

Supplementary file 3. Can the meaning of the colors be described in the file, please.

The colored fields were reformatted with white background.

Supplementary file 5. Can the meaning of the colors be described in the file, please.

The colored fields were reformatted with white background.

Supplementary file 6. Can the meaning of the colors be described in the file, please.

The sentence “The color gradient (green to yellow) highlights the ranking of contig length values” was added to the file and to the file description in the text.

Supplementary file 7. Can the meaning of the colors be described in the file, please.

The colored fields were reformatted with white background.

Supplementary file 8. Can you add the color key again.

The color key was added to the file.